# Association of IL-10 and CRP with Pulse Wave Velocity in Patients with Abdominal Aortic Aneurysm

**DOI:** 10.3390/jcm11051182

**Published:** 2022-02-23

**Authors:** Ida Åström Malm, Rachel De Basso, Peter Blomstrand, Dick Wågsäter

**Affiliations:** 1Department of Natural Sciences and Biomedicine, School of Health and Welfare, Jönköping University, SE-551 11 Jönköping, Sweden; rachel.de.basso@rjl.se (R.D.B.); peter.blomstrand@rjl.se (P.B.); 2Department of Clinical Physiology, County Hospital Ryhov, SE-551 85 Jönköping, Sweden; 3Department of Medical Cell Biology, Uppsala University, SE-751 23 Uppsala, Sweden; dick.wagsater@mcb.uu.se

**Keywords:** arterial stiffness, abdominal aortic aneurysm, diabetes mellitus type 2, inflammatory marker

## Abstract

Background: Markers of inflammation and arterial stiffness are predictors of cardiovascular morbidity and events, but their roles in the mechanisms and progression of abdominal aortic aneurysm (AAA) in males have not been fully investigated. This study explored possible associations between inflammatory marker levels and arterial stiffness in males with AAA. Methods: A total of 270 males (191 AAA and 79 controls) were included in the study. Arterial stiffness was assessed using non-invasive applanation tonometry to measure the regional pulse wave velocity between the carotid and femoral arteries and the carotid and radial arteries. Blood samples were obtained, and interleukin-10 (IL-10) and CRP levels were analysed. Results: Subjects with an AAA had higher levels of IL-10 (21.5 ± 14.0 ng/mL versus 16.6 ± 9.3 ng/mL) compared to controls (*p* = 0.007). In the AAA cohort, subjects with T2DM showed higher levels of IL-10 (26.4 ± 17.3 versus 20.4 ± 13.0, *p* = 0.036). We observed a positive correlation between PWVcf and CRP in the control group (r = 0.332) but not the AAA group. PWVcf and CRP were negatively correlated (r = 0.571) in the T2DM subjects treated with metformin in the AAA group. Conclusion: Arterial stiffness is related to the degree of inflammation reflected by CRP and IL-10 levels in males with an AAA. IL-10 is negatively correlated with arterial stiffness in these subjects. This finding suggests that IL-10 may decrease arterial stiffness in males with AAA. The negative correlation between CRP and PWVcf in males with T2DM treated with metformin may indicate that metformin influences the arterial wall to decrease stiffness in subjects with AAA.

## 1. Introduction

Stiffening in large elastic arteries significantly contributes to cardiovascular disease (CVD) and is a well-known independent predictor of cardiovascular morbidity and mortality [1,2]. Inflammation has a considerable role in the development of arterial stiffness. Macrophages and monocytes generate pro-inflammatory and pro-oxidant cytokines in regions with inflammation. Local inflammation is a protective response of the vascular tissue to eliminate the cause of injury [3,4]. There is an association between arterial stiffness and inflammation measured as C-reactive protein (CRP) levels in healthy individuals [5] and middle-aged and elderly subjects [6], as well as patients with diabetes [7]. Several conditions characterized by increased systemic inflammation are associated with increased arterial stiffness [8,9,10]; for example, abdominal aortic aneurysm (AAA) is a form of CVD characterised by chronic inflammation [11]. Two studies have reported a correlation between increased inflammatory mediator levels and AAA [12,13], and some have found correlations between aneurysm size and inflammatory markers [12,14,15]. Males with AAA have increased arterial stiffness compered to controls [16], indicating that increased arterial stiffness may contribute to the overall higher cardiovascular risk seen in patients with AAA [16].

Immune system activation is related to diabetes incidence and prevalence, and a well-known clinical relationship exists between diabetes, atherosclerosis, and CVD [17]. However, even though type 2 diabetes mellitus (T2DM) is a risk factor for CVD in general, the disease is uncommon in patients with AAA. Indeed, T2DM may protect against AAA through effects on pathophysiological mechanisms and antidiabetic drug treatment [18]. Metformin use in AAA patients is strongly associated with a slower AAA growth rate [19,20,21]. Kunath et al. (2021) demonstrated that metformin reduced vascular contraction and restored the anti-contractile function of perivascular adipose tissue in a mouse model of diabetes [22]. Metformin was also shown to decrease arterial stiffness in patients with non-alcoholic fatty liver disease [23] and polycystic ovary syndrome [24]. Thus, the anti-inflammatory effect of metformin may also influence arterial stiffness. Metformin treatment increases levels of interleukin (IL)-10, an anti-inflammatory cytokine involved in the atherosclerotic process that inhibits both macrophage activation and the expression of matrix metalloproteinases (MMPs), pro-inflammatory cytokines and cyclooxygenase-2 in lipid-loaded and activated macrophage foam cells [25]. Low IL-10 has been associated with an increased risk of future cardiovascular events in patients with acute coronary syndrome [26].

Inflammatory markers and arterial stiffness are predictors of cardiovascular events; however, the roles of these markers in the development and progression of arterial stiffness have not been fully investigated. The aim of the present study was to explore possible associations between inflammatory marker levels and metformin use and arterial stiffness in males with AAA.

## 2. Materials and Methods

### 2.1. Study Population

Participants were recruited from an ongoing ultrasound AAA screening program and a regional ultrasound surveillance program of known AAA in two neighbouring regions in southern Sweden. From 2011–2016, a total of 270 males (mean age 70 ± 4 years) were included in the study: 191 patients with AAA and 79 controls (Table 1). Subjects with the following conditions were excluded: cardiac arrhythmia, severe disability, advanced cancer and language barriers. Subjects with an AAA had a maximum infrarenal aortic diameter of at least 30 mm on their most recent clinical ultrasound examination. Patients with an AAA diameter >55 mm were referred for surgical intervention and excluded from the study. All subjects in the control group had an infrarenal aortic diameter within the reference range (<30 mm) at their screening examinations in the previous 5 years [16].

The regional ethical review board in Linköping, Sweden, approved the study (Dnr 2016/143-32), and all patients accepting gave written consent to participate in the study.

### 2.2. Study Protocol

The participants were instructed to abstain from alcohol for 12 h and from caffeinated beverages and tobacco for 4 h prior to their examination. Examinations were performed at the Department of Clinical Physiology at Linköping University Hospital and Ryhov County Hospital. At the time of the examination, a questionnaire regarding smoking status, CVD, and current medications was completed by the examiner based on participant responses. Arterial stiffness was measured with a SphygmoCor system (Model MM3, AtCor Medical, Sydney, Australia). Simultaneous electrocardiogram recording allowed pulse wave velocity (PWV) to be calculated. The pulse pressure waveforms were non-invasively recorded with a Millar pressure tonometer (Millar, Houston, TX, USA) at the carotid to the femoral PWV (central PWV) and the carotid to the radial artery PWV (peripheral PWV) [16].

### 2.3. Laboratory Analyses

After an overnight fast, morning blood samples were collected in prechilled plastic Vacutainer tubes (Terumo EDTA K-3, Tokyo, Japan). Plasma was prepared by centrifugation at 3000× *g* for 10 min at 4 °C. All samples were stored at −70 °C until analysis in the chemistry laboratories at Linköping University Hospital and Ryhov County Hospital, Jönköping. Both laboratories are ISO/IEC 17025-accredited by the Swedish Board for Accreditation and Conformity Assessment. CRP was analysed by an immunoturbidimetric method according to the manufacturer’s recommendations with ADVIA 1800 (Siemens, Munich, Germany). An inflammatory biomarker panel was analysed with commercial Bio-Plex Pro™ Human Chemokine Panel, 40-Plex kits (Bio-Rad Laboratories, Inc., Hercules, CA, USA) according to the manufacturer’s recommendations. The panel measures a variety of cytokines, chemokines (e.g., eotoxins, Gro, MIP-family), interleukins (e.g., IL-1b, IL-6), interferon gamma, and tumour necrosis factor alpha. An automatic magnetic washer (Magpix, Austin, TX, USA) was used during assay implementation. The 96-well microtiter plates were measured with the Luminex 200 system (Luminex Corp., Austin, TX, USA). A five-parameter logistic curve was generated for each analyte using the Masterplex computer software.

### 2.4. Statistics

Data were analysed using SPSS 27.0 for Windows (IBM Corp., Armonk, NY, USA). Continuous variables are expressed as mean ± standard deviation (SD), and categorical variables are expressed as number of participants and per cent. Comparisons between the AAA and control groups were performed using unpaired Student’s *t*-tests, Fisher’s exact tests, or non-parametric means tests (Mann–Whitney U tests). Pearson’s bivariate correlation coefficients were calculated used to investigate the relationships between PWV and IL-10. Multivariate analysis was made in order to adjust for confounders. Differences were considered significant at *p* ≤ 0.05.

## 3. Results

### 3.1. Demographic Data

Demographic data are presented in Table 1. We saw no differences between AAA and controls regarding height, weight, body mass index, or blood pressure. The AAA cohort was significantly older (*p* < 0.01) compared to the control group. The AAA group also included more current and former smokers (*p* < 0.01). The prevalence rates of T2DM and the reported occurrence of symptomatic cerebrovascular disease were similar between the two groups, while hyperlipidaemia, hypertension, and heart diseases were more frequent in the AAA cohort. A greater proportion of subjects in the AAA population were treated with b-blockers and aspirin (*p* < 0.001). No differences were found in the reported use of other anti-hypertension drugs, including diuretics, calcium channel blockers and statins.

### 3.2. Pulse Wave Parameters, IL-10, and CRP

The AAA cohort had a significantly higher mean level of IL-10 (21.5 ± 14.0 ng/mL) compared to controls (16.6 ± 9.3 ng/mL) (*p* = 0.007). When the AAA cohort was divided into no T2DM (*n* = 156) and T2DM (*n* = 35), the latter subgroup showed a significantly higher level of IL-10 (*p* = 0.036) (Table 2). Among the 35 subjects with T2DM in the AAA group, 12 were treated with metformin and 17 were not, but these subjects had similar IL-10 levels. Several other ILs and chemokines were analysed, but none were significantly correlated with arterial stiffness in the AAA or control group (data not shown) [27].

CRP levels were higher in subjects with AAA compared to controls, 4.5 ± 6.8 mg/L versus 3.0 ± 4.0 mg/L (*p* = 0.018, Table 1). Pearson’s bivariate correlation analyses were performed between central pulse wave velocity (PWVcf) and periphery pulse PWV (PWVcr) and CRP and IL-10 in the AAA and control groups. A significant positive correlation between PWVcf and CRP was found in the controls (r = 0.332, *p* < 0.01) but not in the AAA group. This correlation remained in controls without T2DM (r = 0.359, *p* < 0.01) but disappeared in controls with T2DM. In the AAA group, we found a negative correlation between PWVcf and CRP (r = 0.571, *p* = 0.05), but only in the metformin-treated T2DM subgroup, not in subjects with T2DM without metformin. PWVcr was not correlated with CRP in the control or AAA group. A significant negative correlation was found between PWVcf and IL-10 in subjects with T2DM in the AAA cohort (r = 0.424). However, the significant correlation disappeared after adjusting age, smoking, and blood pressure. The correlation data are summarized in Table 3 and Table 4. No difference was found in PWVcf or PWVcr when we compared AAA subjects with or without T2DM. PWV values were also comparable between AAA subjects with metformin treatment and those without (Table 2). In addition, there was no difference in the control group without AAA for PWV between subjects with or without T2DM.

## 4. Discussion

The main novel finding of our study is the negative correlation between aortic PWV and anti-inflammatory cytokine IL-10 levels in males with AAA. In addition, males with AAA and T2DM had further IL-10 increases compared to subjects with AAA but without T2DM. Interestingly, we found no correlation between aortic and brachial PWV and CRP in the AAA cohort, but these factors were associated with the control group. These data suggest that arterial stiffness is related to inflammation as measured as CRP and IL-10, but this relationship varies in males with and without AAA. This indicates that these inflammatory markers do not affect PWV in males with an AAA, but the results do not exclude the possibility that the vascular inflammation is affected. Further investigations are needed to assess local inflammation and vessel wall morphology.

PWVcf is the gold-standard measurement for arterial stiffness through its proven association with cardiovascular morbidity and mortality [28]. AAA leads to mechanical changes in the aortic wall, which may include a delayed propagation of the pressure wave [29]. It has been suggested that the presence of an aneurysm invalidate the reliability of PWV as a method for arterial stiffness [30]. The speed at which a pulse wave travels through arteries provides a measure of arterial stiffness through the Moens–Kortweg equation [31]. This calculation presumes isotropy within the measured arterial segment, which is not present in AAA, especially in advanced aneurysms. However, it is reasonable to suppose that the structural wall properties along the aorta in the AAA group are far more important for the regional pressure wave propagation speed than the altered geometry in the aneurysmal dilatation [32].

AAA pathogenesis is characterized by chronic inflammation and MMP-mediated aortic wall destruction [11]. Previous studies demonstrated that subjects with AAAs have higher CRP levels [33,34], which was confirmed in our study. CRP is an acute-phase protein that is rapidly produced and released in response to various cellular injuries, making it a useful prognostic marker for CVD [35]. CRP levels are also associated with aneurysm size [33], but this was not found in our study that only included males with small AAAs (<55 mm). CRP levels correlate with PWV-measured arterial stiffness in healthy individuals [5], in line with our results. Nakhai-Pour et al. suggested that elevated CRP may be a consequence of arterial stiffening [6], but we found no association between PWV and CRP in males with AAA. This is surprising because inflammation is an essential feature of AAA and plays an important role in the development of arterial stiffness. However, the AAA cohort was more frequently treated with antihypertensive drugs, which seems to have an impact on low-grade inflammation [36,37]. It is not surprising that a higher percentage in the AAA group were treated with statins, anti-platelet therapy and blood pressure-lowering agents. According to current guidelines should these drugs be considered in all patients with AAA [16,38]. CRP is a strong risk factor for clinical CVD, and arterial stiffness is a known predictor of cardiovascular morbidity and mortality. Interestingly, the relationship between these factors in males with AAA is different compared to other populations. CRP levels should be used with caution to predict CVD in males with AAA. In the subgroup of males with AAA and T2DM who were treated with metformin, we observed a negative correlation between CRP and PWVcf. This suggests that metformin decreases arterial stiffness in subjects with AAA and is in accordance with results in mice showing that metformin reduces vascular contraction [22].

We found higher levels of IL-10 in the AAA cohort compared to controls, and AAA subjects with T2DM had higher levels compared to those without T2DM. IL-10 is an anti-inflammatory cytokine involved in the atherosclerotic process. AAA is characterized by the destruction of elastin, increased collagen levels, and smooth muscle cell apoptosis [39]. Since local inflammation is a protective response of the vascular tissue to eliminate the cause of injury [3,4], the increased levels of IL-10 in subjects with AAA and/or T2DM are not surprising.

The diabetic drug metformin has anti-inflammatory effects [40], but we found no difference in IL-10 levels between T2DM subjects with or without metformin in the AAA cohort. Although metformin can decrease arterial stiffness in several diseases [23,24], these effects are not found in T2DM patients [41]. We found a negative correlation between arterial stiffness and IL-10 levels in AAA subjects with T2DM. However, no difference was found in this correlation between T2DM subjects treated with or without metformin.

The AAA group was not only affected by AAA, but they were also more affected by other CVD. This may not be surprising; AAA shares a number of risk factors in common with other CVD. It was more current and former smokers in the AAA and one of the most well-known risk factors for AAA as well as for CVD is smoking [41].

### Limitations

Our results should be considered in the context of several potential limitations. This was a cross-sectional study, so direct cause-and-effect associations can be derived. The size of the T2DM cohort was small, and even fewer patients were treated with metformin. Moreover, the durations of T2DM and metformin treatment were not assessed. Finally, only males with small AAAs were included in the study. However, the prevalence of AAA is higher in males than females, females are less than one fourth as likely as males to have an AAA.

## 5. Conclusions

Arterial stiffness in males with AAA is related to the level of inflammation as measured by CRP and IL-10 in a different way than in male subjects without AAA. Males with AAA had higher IL-10 levels, which were negatively correlated with arterial stiffness, suggesting that IL-10 may decrease arterial stiffness in males with AAA. We found no relationship between CRP and arterial stiffness in males with AAA, even though they had higher CRP levels and increased arterial stiffness compared to controls. The negative correlation between CRP and PWVcf in males with T2DM who were treated with metformin could indicate that metformin influences the arterial wall and decreases arterial stiffness in subjects with AAA.

## Figures and Tables

**Table 1 jcm-11-01182-t001:** Characteristics of the study population.

	AAA *n* = 191	Controls *n* = 79	*p* Value
Age (year)	70.2 ± 3.0	68.6 ± 3.0	<0.001
Weight (kg)	87.9 ± 13.6	85.0 ± 12.1	0.336
Height (cm)	176.9 ± 6.1	177.6 ± 4.7	0.271
BMI (kg/m^2^)	28.1 ± 4.3	27.2 ± 3.6	0.122
SBP (mmHg)	134 ± 1.4	131 ± 1.9	0.280
DBP (mmHg)	77 ± 0.7	75 ± 2	0.210
CRP (mg/L)	4.5 ± 6.8	3.0 ± 4.0	0.018
Self-reported smoking			
Current *n* (%)	116 (60.7)	26 (32.9)	<0.001
Former *n* (%)	42 (22.0)	6 (7.6)	<0.001
Non-smoker *n* (%)	17 (8.9)	39 (49.5)	<0.001
Self-reported medical treatment			
ACE inhibitors *n* (%)	91 (47.6)	24 (30.4)	0.008
β blockers *n* (%)	84 (44.0)	16 (20.3)	<0.001
Ca antagonists *n* (%)	51 (26.7)	9 (11.4)	0.005
Diuretics *n* (%)	39 (20.4)	5 (6.3)	0.004
Aspirin *n* (%)	117 (61.3)	15 (19.0)	<0.001
Statins *n* (%)	122 (63.9)	3 (3.8)	0.07
Self-reported disease			
HT *n* (%)	153 (80.1)	31 (39.2)	<0.001
HD *n* (%)	73 (38.2)	10 (12.7)	<0.001
T2DM *n* (%)	35 (18.3)	11 (13.9)	0.405
CVD *n* (%)	24 (12.6)	1 (1.3)	0.004
HL *n* (%)	119 (62.3)	17 (21.5)	<0.001

Values are presented as mean ± SD for continuous variables, or as numbers and percentage of participants for categorical variables. *p* values were calculated with t-tests or Mann–Whitney U tests. ACE, angiotensin-converting enzyme; BMI, body mass index; Ca, calcium; CRP, C-reactive protein; CVD, history of symptomatic cerebrovascular disease; DBP: diastolic blood pressure; HD, heart disease; HL, history of hyperlipidaemia; HT, hypertension or taking blood pressure-lowering drugs; SBP: systolic blood pressure; T2DM, type 2 diabetes mellitus.

**Table 2 jcm-11-01182-t002:** PWV, IL-10, T2DM and AAA measurements.

	AAA without T2DM	AAA with T2DM	*p*-Value
	(*n* = 156)	(*n* = 35)	
PWVcf (m/s)	12.1 ± 2.8	13.2 ± 3.4	0.068
PWVcr (m/s)	9.5 ± 1.3	9.2 ± 1.2	0.259
IL-10 (ng/mL)	20.4 ± 13.0	26.4 ± 17.3	0.036
CRP (mg/L)	4.3 ± 6.3	5.2 ± 8.7	0.774
	**AAA with T2DM and Metformin**	**AAA with T2DM without Metformin**	** *p* ** **-Value**
	(*n* = 12)	(*n* = 17)	
PWVcf (m/s)	12.8 ± 3.8	13.5 ± 3.2	0.565
PWVcr (m/s)	9.0 ± 1.4	9.3 ± 1.0	0.205
IL-10 (ng/mL)	29.9 ± 21.4	24.2 ± 14.3	0.468
CRP (mg/L)	6.5 ± 11.0	3.2 ± 2.9	0.875

Values are presented as mean ± SD. AAA, abdominal aortic aneurysm; CRP, C-reactive protein; IL-10, interleukin 10; PWVcf, carotid femoral pulse wave velocity; PWVcr, carotid-radial pulse wave velocity; T2DM, type 2 diabetes mellitus.

**Table 3 jcm-11-01182-t003:** Correlations between PWV and CRP.

PWVcf (m/s)	R	PWVcr (m/s)	R
All AAA	−0.080		−0.083
AAA without T2DM	−0.061		−0.093
AAA with T2DM	−0.142		−0.055
AAA with T2DM and metformin	−0.571 **		−0.122
AAA with T2DM without metformin	−0.106		−0.122
All control	0.332 *		−0.173
Control with T2DM	0.094		−0.296
Control without T2DM	0.359 *		−0.182

Pearson’s variate correlation, * *p* < 0.01, ** *p* = 0.05. AAA, abdominal aortic aneurysm; PWVcf, carotid femoral pulse wave velocity; PWVcr, carotid-radial pulse wave velocity; T2DM, type 2 diabetes mellitus.

**Table 4 jcm-11-01182-t004:** Correlations between PWV and IL-10.

PWVcf (m/s)	R	PWVcr (m/s)	R
All AAA	−0.206 *		−0.233 *
AAA without T2DM	−0.181 *		−0.232 *
AAA with T2DM	−0.424 *		−0.143
AAA with T2DM and metformin	−0.433		−0.059
AAA with T2DM without metformin	−0.408		−0.244
All control	0.008		0.001
Control with T2DM	−0.012		0.589
Control without T2DM	−0.016		−0.094

Pearson’s bivariate correlation, * *p* < 0.05. AAA, abdominal aortic aneurysm; IL-10, interleukin 10; PWVcf, carotid femoral pulse wave velocity; PWVcr, carotid-radial pulse wave velocity; T2DM, type 2 diabetes mellitus.

## Data Availability

The data supporting reported results can be found at Department of Natural Sciences and Biomedicine, School of Health and Welfare, Jönköping University, Jönköping, Sweden.

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
