# Peer review of "Association of IL-10 and CRP with Pulse Wave Velocity in Patients with Abdominal Aortic Aneurysm"

_jcm, 2022, doi:10.3390/jcm11051182_

Round 1
Reviewer 1 Report
This study presents the vascular protective role of IL-10. An elevated level of IL-10 is associated with a low risk of inflammation in the vascular wall. The study was performed in male subjects in whom the frequency of abdominal aortic aneurysm is higher. In subjects with type 2 diabetes mellitus treated with Metformin, included in the study, elevated IL-10 levels were associated with a significant vascular protective effect. The authors found a negative correlation between arterial stiffness and IL -10 levels in abdominal aortic aneurysm subjects and with type 2 diabetes mellitus in treatment with Metformin. I consider the information somehow useful for medical practice because it has several limitations: cross sectional study, only males with small AAA, small T2DM cohort. The authors should also have to consider that the arterial pulse wave decreases in elderly. The article is well written and documented.
Author Response
Dear Editor and reviewers,
First of all, we are thankful for the positive response on our manuscript and the comments raised. We have answered all the comments and made clarifications in the manuscript.
Reviewer 1
This study presents the vascular protective role of IL-10. An elevated level of IL-10 is associated with a low risk of inflammation in the vascular wall. The study was performed in male subjects in whom the frequency of abdominal aortic aneurysm is higher. In subjects with type 2 diabetes mellitus treated with Metformin, included in the study, elevated IL-10 levels were associated with a significant vascular protective effect. The authors found a negative correlation between arterial stiffness and IL -10 levels in abdominal aortic aneurysm subjects and with type 2 diabetes mellitus in treatment with Metformin. I consider the information somehow useful for medical practice because it has several limitations: cross sectional study, only males with small AAA, small T2DM cohort. The authors should also have to consider that the arterial pulse wave decreases in elderly. The article is well written and documented.
I consider the information somehow useful for medical practice because it has several limitations: cross sectional study, only males with small AAA, small T2DM cohort.
Answer: Thank you for your comments. We agree that the study has several limitations; cross-sectional design, only males with small AAA are included, and also a small T2DM cohort. These limitations are stated in section 4.1. According to Swedish guidelines, patients with an AAA diameter >55 mm are referred for surgical intervention and were therefore difficult to include in the study.
Page 11, line 246
Page 2, line 82-83
The authors should also have to consider that the arterial pulse wave decreases in elderly.
Answer: Thank you for your comments. However, the central arterial stiffness increase with age which result in an increase in pulse wave velocity. The AAA group and the control group are close to each other in age (70 years vs 69 years). Thus, the difference may not be due to age but rather to AAA disease.

Reviewer 2 Report
Ida Åström Malm et al reported a negative correlation between aortic PWV and IL-10 in patients with AAA. Furthermore, they shown an increase of IL-10 in AAA with T2DM compared to subjects without T2DM. They confirm the positive correlation between PWVcf and CRP in the control group however, no correlation was found in AAA group. In addition PWVcf and CRP were negatively correlated in the T2DM subjects treated with metformin in the AAA group. The study is interesting, however several limitations may decrease the scientific impact and to conclude robustly such results.
- Number of subjects especially in AAA subgroups
- Only male subjects were used for the study which may hide the reel effect in presence of female group. And if the effects are male-dependent (which could be the main originality of the paper), the number is low for such conclusion.
- Using cytokine markers in single time point is tricky and confusing for some results to interpret as anti-inflammatory or compensatory effect.
Minor comments
Some typo error e.i line 192 “in for arterial”
Line 220-2022 Ref
Author Response
Dear Editor and reviewers,
First of all, we are thankful for the positive response on our manuscript and the comments raised. We have answered all the comments and made clarifications in the manuscript.
Ida Åström Malm et al reported a negative correlation between aortic PWV and IL-10 in patients with AAA. Furthermore, they shown an increase of IL-10 in AAA with T2DM compared to subjects without T2DM. They confirm the positive correlation between PWVcf and CRP in the control group however, no correlation was found in AAA group. In addition PWVcf and CRP were negatively correlated in the T2DM subjects treated with metformin in the AAA group. The study is interesting, however several limitations may decrease the scientific impact and to conclude robustly such results.
Number of subjects especially in AAA subgroups
Answer: Thank you for your comment. Diabetes has a protective role on the development of AAA, which unfortunately affects the number of subjects with diabetes and metformin treatment in the AAA-group. This study should be considered as a pilot study for larger follow ups in order to include more of the more rare subgroups. Since puls wave measurments on AAA patients is not included in standard diagnostic it was difficult for us to find cohorts where this was analysed. We therefore performed the analysis on the available patients since we find the results of importance that can explain mechanisms of metformin on vascular function in patients with AAA.
Shantikumar S, Ajjan R, Porter KE, Scott DJA. Diabetes and the Abdominal Aortic Aneurysm. European Journal of Vascular and Endovascular Surgery. 2010;39(2):200-7.
Yuan Z, Heng Z, Lu Y, Wei J, Cai Z. The Protective Effect of Metformin on Abdominal Aortic Aneurysm: A Systematic Review and Meta-Analysis. Frontiers in Endocrinology. 2021;12.
Only male subjects were used for the study which may hide the reel effect in presence of female group. And if the effects are male-dependent (which could be the main originality of the paper), the number is low for such conclusion.
Answer: We agree with the reviewer, it is a limitation that only males are included in the study, and this is stated in the section of limitation. However, the prevalence of AAA is higher in males than females, females are less than one fourth as likely as males to have an AAA. The statistical power would be to small if females would be included.
Page 11, Line 254
Using cytokine markers in single time point is tricky and confusing for some results to interpret as anti-inflammatory or compensatory effect.
Answer: We agree with this, nevertheless the results clearly shows a significant association for CRP and IL-10 and stiffness under these circumstances and reflects these factors as potential markers. Experimental studies using IL-10 treatment in mice support this, strengthening our results.
Sikka G, Miller KL, Steppan J, Pandey D, Jung SM, Fraser CD 3rd, Ellis C, Ross D, Vandegaer K, Bedja D, Gabrielson K, Walston JD, Berkowitz DE, Barouch LA. Interleukin 10 knockout frail mice develop cardiac and vascular dysfunction with increased age. Exp Gerontol. 2013 Feb;48(2):128-35. doi: 10.1016/j.exger.2012.11.001.
Minor comments
Some typo error e.i line 192 “in for arterial”
Line 220-2022 Ref
Answer: Thank you for notice this. This is now change in the manus.

Round 2
Reviewer 2 Report
authors responses are clear and convincing